# Impact of COVID-19 Vaccination on Cardiac Function and Survival in Maintenance Hemodialysis Patients

**DOI:** 10.3390/vaccines13030208

**Published:** 2025-02-20

**Authors:** Xiao Tu, Tingfei He, Bing Xu, Jiazhen Yin, Fangyu Yi, Ye Li, Jinchi Gao, Peng Bi, Wanyue Xu, Rihong Hu, Lidan Hu, Yayu Li

**Affiliations:** 1Key Laboratory of Kidney Disease Prevention and Control Technology, Department of Nephrology, Hangzhou TCM Hospital of Zhejiang Chinese Medical University (Hangzhou Hospital of Traditional Chinese Medicine), 453 Tiyu Road, Xihu District, Hangzhou 310012, China; tuxiao0819@126.com (X.T.); yinjiazhen@163.com (J.Y.); 0618107@zju.edu.cn (P.B.); xuwanyue73@163.com (W.X.); 2Hangzhou Clinical College, Zhejiang Chinese Medical University, 548 Binwen Road, Hangzhou 310053, China; s1137072939@163.com (T.H.); xubinggogogo@163.com (B.X.); apackmed@163.com (F.Y.); taeyeon30935@gmail.com (Y.L.); gjc19857169066@163.com (J.G.); 3Hemodialysis Unit, Hangzhou Hospital of Traditional Chinese Medicine, 453 Tiyu Road, Xihu District, Hangzhou 310012, China; hzhurihong@163.com; 4Department of Nephrology, The Children’s Hospital, Zhejiang University School of Medicine, National Clinical Research Center for Child Health, 3333 Binsheng Road, Binjiang District, Hangzhou 310003, China

**Keywords:** COVID-19 vaccination, maintenance hemodialysis, cardiac function, echocardiography, BNP, survival rate

## Abstract

Maintenance hemodialysis patients are at increased risk of cardiovascular complications and mortality following COVID-19 infection due to compromised immune function. This study aims to evaluate the impact of the COVID-19 vaccine (CoronaVac) on cardiac function and survival in this population. **Background/Objectives**: We aimed to examine whether CoronaVac vaccination affects heart function and survival rates in maintenance hemodialysis patients. Specifically, we assessed changes in heart ultrasound (echocardiographic) measurements, B-type natriuretic peptide (BNP) levels, and survival outcomes by comparing vaccinated and unvaccinated patients. **Methods**: A retrospective analysis was conducted on 531 maintenance hemodialysis patients, including 79 who received CoronaVac and 452 who did not. We compared the pre- and post-infection changes in heart function (echocardiographic parameters) and BNP levels between the two groups and assessed their association with the survival rates. **Results**: The vaccinated patients were younger (60.54  ±  13.51 vs. 65.21  ±  13.76 years, *p* = 0.006) and had shorter dialysis durations (56.04  ±  51.88 vs. 73.73  ±  64.79 months, *p* = 0.022). The mortality rate was also significantly lower in the vaccinated group (6.33% vs. 14.38%, *p* = 0.049). After infection, the unvaccinated patients showed significant declines in heart function and increased B-type natriuretic peptide levels, while the vaccinated patients demonstrated no significant deterioration. Older age, coronary artery disease, inflammation levels, and heart abnormalities were identified as the key risk factors for mortality. **Conclusions**: CoronaVac was linked to lower mortality and better heart function in maintenance hemodialysis patients. The vaccine may help to reduce infection severity, lower strain on the heart, and improve the overall prognosis.

## 1. Introduction

Since its initial outbreak in 2019, COVID-19 has rapidly spread, becoming a major global public health challenge. Hemodialysis patients, due to their immunocompromised status and multiple comorbidities, are particularly vulnerable to COVID-19 infection. An infection rate of 29.31% was reported among dialysis patients, primarily due to frequent interactions with healthcare facilities and increased exposure to high-risk environments [1]. Similarly, high infection rates in European dialysis centers have been linked to close contact within these facilities [2].

COVID-19 has significantly impacted dialysis patients, with the mortality rates reaching 30.5%, and early lymphopenia and elevated lactate dehydrogenase were identified as predictors of poor outcomes [3]. The mortality in elderly dialysis patients was reported to be six times higher than in younger patients [4]. Additionally, mortality rates of 33.82% and 39.08% were documented in separate studies, highlighting the vulnerability of this population [5]. Frequent healthcare exposure and treatment interruptions, including disruptions in dialysis due to worsening conditions, have further complicated patient management during the pandemic. Risk factors such as chronic pulmonary and cardiovascular diseases and elevated C-reactive protein (CRP) and D-dimer levels have also been linked to poorer outcomes following COVID-19 infection [6].

Vaccination has been shown to provide substantial protection for dialysis patients, a highly vulnerable population. Two doses of the vaccine were reported to reduce their hospitalization rates by 75% and mortality by 88% [7]. Additionally, vaccination offered significant protection against variants such as Omicron, demonstrating its efficacy even in the context of emerging viral strains [8]. However, concerns regarding vaccine safety in hemodialysis patients persist given their immunocompromised status and multiple comorbidities. This population may be more prone to adverse effects or rare complications, including severe allergic reactions and localized infections. While the benefits of vaccination, such as reduced hospitalizations and mortality, are well documented [8], addressing these safety concerns remains crucial to optimizing the vaccine strategies in this high-risk group.

Cardiovascular disease is highly prevalent among dialysis patients, with incidence rates between 40% and 60%, making it a leading cause of mortality [9]. The mechanisms underlying impaired cardiac function include volume overload, metabolic disturbances, and oxidative stress [10]. Undergoing maintenance hemodialysis (MHD) further exacerbates these issues as repeated fluid shifts and metabolic imbalances can place additional strain on the heart. Routine assessments using echocardiography and biomarkers (e.g., BNP or NT-proBNP), along with optimized dialysis strategies, metabolic correction, and appropriate pharmacological intervention (e.g., ACE inhibitors or beta-blockers), can effectively improve the prognosis [11]. Conditions like left ventricular hypertrophy and reduced ejection fraction are independent risk factors for cardiovascular events and all-cause mortality, emphasizing the critical importance of cardiac health in dialysis patients [12].

COVID-19 may also result in long-term cardiac sequelae. Studies have shown that 78% of recovered patients exhibit myocardial involvement, with 60% having persistent myocarditis [13]. In the MHD population, these chronic cardiac effects may be further magnified by pre-existing fluid overload and systemic inflammation, potentially accelerating cardiac deterioration. Patients may experience arrhythmias and thrombosis due to systemic inflammation and hypercoagulability triggered by the virus [14]. Vaccination, in addition to reducing the acute-phase infection risk, also lowers the incidence of long-term cardiovascular complications. The data suggest that patients receiving two vaccine doses displayed a 30% reduction in the risk of long-term cardiovascular complications [15]. Furthermore, vaccination reduced the mortality and cardiovascular event rates by 91% among patients with underlying heart conditions, providing crucial protection for high-risk groups [16]. Vaccination also notably decreased the occurrence of post-COVID sequelae, such as hypertension and heart failure [17].

CoronaVac is a whole inactivated virus COVID-19 vaccine, CoronaVac (also called Sinovac COVID-19) and was created by Sinovac Biotech, a Chinese company. Studies specifically evaluating the effect of the CoronaVac vaccine on cardiac function in dialysis patients are limited. Akin et al. [18] reported that, while the immune response was weaker among hemodialysis patients receiving CoronaVac [19], the antibody levels were still sufficient for protective immunity. Medina-Pestana et al. [20] showed that, although the breakthrough infection rates were relatively high among dialysis patients post-CoronaVac, the hospitalization and mortality rates were significantly reduced. Ran et al. [21] documented a case of IgA nephropathy exacerbation leading to dialysis following CoronaVac vaccination, indicating the need for close renal function monitoring. Ihara et al. [22] observed that the administration of the 23-valent pneumococcal polysaccharide vaccine significantly reduced cardiovascular events and improved the survival among dialysis patients, especially when co-administered with the influenza vaccine.

During the Omicron variant outbreak, vaccination continued to demonstrate significant protective effects, reducing the hospitalization risk among dialysis patients who completed vaccination or received booster doses by 59%. Nevertheless, concerns persist regarding adverse reactions following vaccination, particularly related to cardiac function. Specifically, hypertensive patients were more likely to develop mild local or systemic reactions post-CoronaVac, although the overall safety profile remained favorable [23]. Additionally, comparative research has found that, while inactivated vaccines have lower seroconversion rates compared to mRNA vaccines, they are better tolerated, especially in immunosuppressed individuals [24].

Despite these findings, direct evidence on how CoronaVac affects cardiac function in MHD patients remains scarce. As MHD patients already face elevated cardiovascular risks, understanding whether the vaccine aggravates cardiac impairment or provides protective benefits is essential. Randomized controlled trials are necessary to clarify the specific impact of CoronaVac on the cardiovascular outcomes in this population, thereby informing the clinical practice and guiding the vaccination strategies in this high-risk group.

## 2. Materials and Methods

### 2.1. Study Population

This retrospective study was conducted at the Hemodialysis Unit of Hangzhou Traditional Chinese Medicine Hospital in November 2022, following the Declaration of Helsinki and approved by the hospital’s Ethics Committee (approval number: 2023KLL027). Written informed consent was obtained from all participants. A total of 1028 chronic maintenance hemodialysis patients were screened. Inclusion criteria: all patients who were undergoing maintenance hemodialysis during the COVID-19 pandemic at Hangzhou Traditional Chinese Medicine Hospital. Exclusion criteria: patients were excluded if they had died before the COVID-19 pandemic (n = 3), transferred to other hospitals (n = 383), transitioned to peritoneal dialysis (n = 22), underwent renal transplant (n = 13), experienced renal function recovery (n = 20), or were lost to follow-up (n = 37). This resulted in 587 eligible patients, of whom 56 were further excluded due to missing echocardiographic data, leaving 531 participants for analysis. Participants were categorized into vaccinated (CoronaVac, n = 79) and unvaccinated groups (n = 452). See Figure 1 for details.

### 2.2. Data Collection

Clinical data were retrieved from the Hospital Information System (HIS), including vaccination status (primary, booster, and secondary booster doses), demographic characteristics (age, gender, and comorbidities), and COVID-19 outcomes (symptomatic infections, hospitalizations, COVID-19-related deaths, and all-cause mortality). Clinical indicators included echocardiographic parameters (e.g., interventricular septal thickness, left ventricular internal diameters in diastole and systole, posterior wall thickness, aortic root dimension, left atrial diameter, stroke volume, ejection fraction, and fractional shortening), which were classified as normal or abnormal. Additional markers included brain natriuretic peptide (BNP; maximum and mean levels), biochemical markers (e.g., albumin, creatinine, blood urea nitrogen, lipid profiles, and liver enzymes), hemoglobin, inflammatory markers (e.g., high-sensitivity CRP and ferritin), mineral metabolism indices (e.g., parathyroid hormone, calcium, and phosphate), and dialysis adequacy metrics (e.g., KT/V and urea reduction ratio). Data were tracked longitudinally from November 2020 to November 2024 to evaluate trends and outcomes.

### 2.3. Statistical Methods

All analyses were conducted using SPSS version 26.0. Continuous variables were reported as mean ± standard deviation (SD) or median (Q1, Q3) depending on the data distribution, while categorical variables were expressed as frequencies and percentages. Group comparisons were conducted using appropriate statistical tests (e.g., *t*-test, Mann–Whitney U test, and chi-squared test). Survival analysis was performed using Cox proportional hazard models, with hazard ratios (HRs) and 95% confidence intervals (CIs) reported. A *p*-value < 0.05 was considered statistically significant.

## 3. Results

### 3.1. Baseline Characteristics of the Study Population

This study included 531 maintenance hemodialysis patients, of whom 79 were vaccinated with CoronaVac and 452 were unvaccinated. The vaccinated group was younger (60.5 vs. 65.2 years, *p* = 0.006) and had a shorter dialysis duration (56.0 vs. 73.7 months, *p* = 0.022). Chronic glomerulonephritis and diabetic nephropathy were the most common causes of kidney failure. The vaccinated group had a significantly lower prevalence of coronary artery disease (10.1% vs. 29.0%, *p* < 0.001). The details are presented in Table 1.

### 3.2. Clinical and Biochemical Differences Between Survival and Death Groups

The vaccinated group had a significantly lower mortality rate (6.3% vs. 14.4%, *p* = 0.049), suggesting a potential protective effect. The multivariate analysis identified older age (HR: 1.10, *p* = 0.009), coronary artery disease (HR: 4.20, *p* = 0.041), and inflammation levels (hs-CRP, HR: 1.03, *p* = 0.049) as independent mortality predictors. Notably, abnormal left ventricular systolic diameter (LVIDs) was also a strong independent predictor (HR: 5.60, *p* = 0.018). The details are presented in Table 2.

### 3.3. Echocardiographic and BNP Characteristics of the Study Population

The echocardiographic parameters and BNP levels were analyzed for 531 maintenance hemodialysis patients. The BNP levels, a marker of cardiac stress, rose significantly post-COVID-19 in the unvaccinated patients (*p* < 0.001) but remained stable in the vaccinated patients (*p* > 0.05), suggesting a protective effect against infection-related cardiac strain.

The echocardiographic differences were limited. The interventricular septal thickness (IVSD) was slightly higher in the death group, but this was not statistically significant (*p* = 0.165). Among the echocardiographic parameters, only abnormal left ventricular end-systolic diameter (LVIDs) showed a significant association with mortality (20.00% vs. 7.83%, *p* = 0.049). Coronary artery disease (67.14% vs. 19.96%, *p* < 0.001) and heart failure (11.43% vs. 3.47%, *p* = 0.007) were also more prevalent in the death group.

These findings suggest that elevated BNP levels and coronary artery disease are key predictors of mortality, with left ventricular dysfunction as a potential contributing factor. The details are presented in Table 3.

### 3.4. Multivariate Analysis of LVIDs and Related Factors

A multivariate Cox regression analysis was performed to explore the association between LVIDs and survival outcomes while adjusting for confounding variables. The results revealed that the patients with abnormal LVIDs had significantly higher HRs for mortality compared to those with normal LVIDs (HR: 5.60, 95% CI: 1.34–23.34, *p* = 0.018) (see Table 4). This suggests that abnormal LVIDs is an independent predictor of mortality in the maintenance hemodialysis population.

Among the clinical parameters, age was identified as another significant predictor, with each one-year increase in age associated with a 10% higher risk of mortality (HR: 1.10, 95% CI: 1.02–1.18, *p* = 0.009). Elevated Hs-CRP was also independently associated with increased mortality risk (HR: 1.03, 95% CI: 1.01–1.05, *p* = 0.049), reflecting the role of systemic inflammation in adverse outcomes. Coronary artery disease (CAD) was a strong independent risk factor, with an HR of 4.20 (95% CI: 1.06–16.71, *p* = 0.041), indicating its significant impact on mortality.

Conversely, other variables such as albumin (ALB), serum creatinine (Cr), phosphate (PHOS), urea, prothrombin time (PT), prealbumin (PA), and heart failure (HF) did not reach statistical significance in the multivariate model. Interestingly, both BNP Max and BNP Mean, although significant in the univariate analysis, were not independent predictors in the multivariate analysis after adjusting for other factors.

These findings emphasize the importance of echocardiographic parameters, particularly LVIDs, and the underlying systemic factors like age, inflammation, and CAD in predicting mortality in hemodialysis patients. This highlights the need for close monitoring and targeted interventions to address these risk factors in clinical practice.

**Table 4 vaccines-13-00208-t004:** Multivariate Cox regression analysis of predictors of mortality in maintenance hemodialysis patients.

Univariate						Multivariate				
Variable	β	S.E	Z	*p*-Value	HR (95% CI)	Variable	β	S.E	Z	*p*-Value
Age	0.07	0.01	7.23	<0.001 **	1.08 (1.06–1.10)	0.1	0.04	2.6	0.009 **	1.10 (1.02–1.18)
ALB	−0.11	0.02	−4.48	<0.001 **	0.90 (0.85–0.94)	0.13	0.12	1.05	0.294	1.14 (0.89–1.45)
Cr	−0.01	0	−4.3	<0.001 **	0.99 (0.99–0.99)	0	0	0.12	0.906	1.00 (1.00–1.00)
PHOS	−1.16	0.3	−3.86	<0.001 **	0.31 (0.17–0.57)	−0.56	0.93	−0.6	0.547	0.57 (0.09–3.54)
Urea	−0.07	0.02	−3.34	<0.001 **	0.93 (0.89–0.97)	−0.09	0.07	−1.3	0.195	0.91 (0.79–1.05)
PT	0.06	0.03	2.2	0.028 *	1.07 (1.01–1.13)	−0.03	0.17	−0.16	0.869	0.97 (0.69–1.37)
PA	−0.01	0	−3.29	<0.001 **	0.99 (0.99–0.99)	0	0.01	0.01	0.989	1.00 (0.99–1.00)
AST	0.04	0.01	3.54	<0.001 **	1.04 (1.02–1.06)	0.03	0.02	1.43	0.154	1.03 (0.99–1.08)
Hb	−0.03	0.01	−3.5	<0.001 **	0.97 (0.96–0.99)	−0.04	0.03	−1.34	0.18	0.96 (0.90–1.02)
UA	−0.01	0	−4.52	<0.001 **	0.99 (0.99–0.99)	0	0	0.55	0.584	1.00 (0.99–1.01)
Hs-CRP	0.01	0	4.84	<0.001 **	1.01 (1.01–1.02)	0.03	0.01	1.97	0.049 *	1.03 (1.01–1.05)
BNP Max	0.01	0	4.93	<0.001 **	1.01 (1.01–1.01)	0	0	−0.19	0.847	1.00 (1.00–1.00)
BNP Mean	0.01	0	2.1	0.036 *	1.01 (1.01–1.01)	0	0	−0.66	0.506	1.00 (1.00–1.00)
HTN	−0.81	0.28	−2.92	0.004 **	0.44 (0.26–0.77)	−0.63	0.73	−0.86	0.388	0.53 (0.13–2.23)
CAD	1.93	0.25	7.57	<0.001 **	6.88 (4.18–11.34)	1.44	0.7	2.04	0.041 *	4.20 (1.06–16.71)
HF	1.17	0.38	3.12	0.002 **	3.22 (1.54–6.74)	−0.18	1.37	−0.13	0.896	0.84 (0.06–12.34)

Abbreviations: ALB: albumin; Cr: creatinine; PHOS: phosphate; Urea: urea nitrogen; PT: prothrombin time; PA: prealbumin; AST: aspartate transaminase; Hb: hemoglobin; UA: uric acid; Hs-CRP: high-sensitivity; BNP Max: brain natriuretic peptide maximum; BNP Mean: brain natriuretic peptide mean; HTN: hypertension; CAD: coronary artery disease; HF: heart failure; HR: hazard ratio; CI: confidence interval; S.E: standard error. * *p* < 0.05, ** *p* < 0.01.

### 3.5. Multivariate Cox Regression Analysis Across Different Models

To evaluate the relationship between echocardiographic parameters and mortality in hemodialysis patients, three Cox regression models were developed. Model 1 provided crude (unadjusted) HRs, while Model 2 accounted for hypertension (HTN), arrhythmia, CAD, and heart failure (HF). Model 3 further adjusted for age, HTN, arrhythmia, CAD, HF, albumin (ALB), and prealbumin (PA). The results demonstrated the significance of specific cardiac parameters in predicting mortality.

Among the echocardiographic parameters, LVIDs1 was consistently identified as an independent predictor of mortality across all the models. In the unadjusted analysis (Model 1), an abnormal LVIDs1 was associated with a twofold higher risk of mortality (HR: 2.66, 95% CI: 1.16–6.10, *p* = 0.021). After adjusting for additional variables in Model 2, the risk increased significantly (HR: 28.88, 95% CI: 3.47–240.13, *p* = 0.002). Model 3, which included demographic and nutritional variables, further amplified this association (HR: 1574.03, 95% CI: 3.30–750,000.88, *p* = 0.019), underscoring the importance of LVIDs1 as a robust marker of poor prognosis.

Gender also emerged as a significant factor in the adjusted models. Women demonstrated a substantially reduced risk of mortality compared to men, with HRs of 0.15 (95% CI: 0.03–0.66, *p* = 0.013) and 0.01 (95% CI: 0.00–0.34, *p* = 0.011) in Models 2 and 3, respectively. Additionally, left ventricular end-diastolic diameter (LVIDd1) and ejection fraction (EF1) became significant predictors in Models 2 and 3, highlighting their importance in outcomes when confounding factors were considered.

Other parameters, including interventricular septal diameter (IVSD1), stroke volume (SV1), and aortic diameter (AO1), showed relevance in the adjusted models, reflecting their potential role in determining survival. However, some variables, such as left ventricular posterior wall diameter (LVPWD1) and fractional shortening (FS1), did not exhibit consistent significance across all the models.

Overall, the findings highlight the critical role of echocardiographic parameters, particularly LVIDs1, in predicting mortality risk in the hemodialysis population. The strong associations observed after adjusting for clinical and demographic confounders emphasize the need for regular cardiac assessments and targeted interventions to improve patient outcomes (see Table 5).

### 3.6. BNP Levels and the Effect of Vaccination Pre- and Post-Infection in Hemodialysis Patients

This study evaluated the impact of COVID-19 infection on the BNP levels in maintenance hemodialysis patients, focusing on the influence of vaccination. Among the unvaccinated patients, the BNP levels significantly increased post-infection, with BNP Max showing a notable rise (*p* < 0.001) but only a moderate shift according to the z-value analysis (*p* = 0.078). Similarly, the BNP Mean levels increased significantly (*p* < 0.001), although the z-value analysis indicated a less pronounced change (*p* = 0.119). In the vaccinated patients, the post-infection BNP Max levels exhibited a decrease that was not statistically significant by a Wilcoxon test (*p* = 0.3). However, the z-value analysis revealed a significant reduction (*p* = 0.006). The BNP Mean levels also rose in the vaccinated group, but this increase was not significant (*p* = 0.098) regarding the Wilcoxon test, while the z-value analysis indicated a significant difference (*p* = 0.02). These findings suggest that vaccination may play a role in moderating changes in BNP levels following COVID-19 infection. Detailed data are presented in Table 6 and Figure 2.

### 3.7. Impact of Vaccination on Echocardiographic Parameters

This study investigated changes in echocardiographic parameters before and after COVID-19 infection in maintenance hemodialysis patients, comparing those vaccinated with CoronaVac to those who were unvaccinated. The unvaccinated patients showed significant post-infection deterioration in multiple echocardiographic parameters, including increased ventricular wall thickness (IVSD, *p* < 0.001; LVPWD, *p* = 0.001), reduced cardiac efficiency (EF, *p* = 0.002), and impaired heart contraction (FS, *p* = 0.017). In contrast, no significant structural or functional decline was observed in the vaccinated patients (all *p* > 0.05). For example, IVSD and LVPWD remained stable (*p* = 0.423 and *p* = 0.934), and EF and FS showed no significant decline (*p* = 0.971 and *p* = 0.810). These findings suggest that vaccination may help to preserve heart function after infection. Detailed data are presented in Table 7 and Figure 3.

## 4. Discussion

As shown in Table 1, significant differences were observed in the baseline characteristics of the maintenance hemodialysis patients based on their COVID-19 vaccination status. The vaccinated patients were younger (60.54 ± 13.51 vs. 65.21 ± 13.76 years, *p* = 0.006) and had shorter dialysis durations (56.04 ± 51.88 vs. 73.73 ± 64.79 months, *p* = 0.022). The prevalence of CAD and the mortality rate were significantly lower in the vaccinated patients (10.13% vs. 28.98%, *p* < 0.001; 6.33% vs. 14.38%, *p* = 0.049). These findings suggest that younger, healthier individuals were more likely to receive the COVID-19 vaccine, reflecting better health awareness. This trend aligns with prior research indicating that younger and healthier individuals are more inclined to accept vaccination [25]. Instead, it likely reflects a significant bias, where patients with pre-existing cardiovascular conditions may have been less likely to receive the vaccine due to concerns about their overall health status. The burden of CAD in the vaccinated group might have a considerable influence on the study results and warrants further exploration. In dialysis patients, vaccination appears to correlate with better overall health, as evidenced by the lower systemic inflammation and stronger immune responses—factors that are associated with improved cardiovascular health [26]. The observed differences suggest that patients in better health are more likely to choose vaccination, a pattern that is consistent with previous findings [27].

The patients in the mortality group had significantly poorer nutritional, metabolic, and inflammatory markers compared to the survivors (Table 2) and a higher prevalence of cardiovascular complications, consistent with studies on age, malnutrition, and inflammation as key risk factors in dialysis mortality [28,29,30]. Age emerged as a significant risk factor, with the mortality group being considerably older (75.67 ± 12.27 vs. 62.82 ± 13.24 years, *p* < 0.001). This finding aligns with the observation that older patients with CKD face elevated risks of cardiovascular events, malnutrition, and immunosuppression. Older patients typically encounter more complications, such as cardiovascular diseases, malnutrition, and impaired immune function, which collectively contribute to their significantly increased mortality risk. Age is widely recognized as an independent predictor of increased mortality in this population [31]. In terms of nutritional and metabolic markers, the mortality group had significantly lower albumin levels, a well-known indicator of nutritional status, which correlates with mortality risk in CKD patients [28,32]. Similarly, the creatinine levels were lower in the mortality group, suggesting severe muscle wasting or malnutrition [33]. The hemoglobin and phosphorus levels were also reduced in the mortality group, indicating inadequate protein intake and potential metabolic disturbances [29,34,35]. The inflammatory markers, particularly high-sensitivity Hs-CRP, were significantly elevated in the mortality group, reflecting heightened systemic inflammation [29,36]. Hs-CRP is a sensitive biomarker of inflammation and is strongly associated with cardiovascular events and mortality [32,36]. Additionally, PT was prolonged in the mortality group, potentially indicating impaired liver function, malnutrition, or comorbid conditions such as heart failure [34]. The interplay between malnutrition and inflammation is well documented in dialysis patients, often forming the “malnutrition-inflammation-atherosclerosis (MIA) syndrome”. This synergistic mechanism significantly elevates the risk of cardiovascular events and all-cause mortality [35,37]. This mechanism is especially important in older dialysis patients as aging exacerbates the negative effects of chronic inflammation and malnutrition, further increasing the risk of mortality.

This study demonstrates significant differences in the BNP levels and echocardiographic parameters between the survival and death groups of dialysis patients. Specifically, the BNP levels were markedly higher in the death group, and the incidence of cardiovascular complications, such as coronary artery disease (CAD) and heart failure (HF), was significantly higher, along with a higher abnormality rate of LVIDs. The importance of BNP and LVIDs as key markers of cardiovascular events in dialysis patients has been further validated. These findings are supported and extended by previous studies. Palazzuoli [38] found that BNP levels were closely associated with left ventricular mass, volume, and function in dialysis patients, particularly in relation to left ventricular enlargement and diastolic dysfunction caused by volume overload in CKD patients. Zoccali [39] further emphasized that BNP levels are independent predictors of all-cause and cardiovascular mortality in dialysis patients, correlating with left ventricular mass and function, suggesting that BNP could be a valuable biomarker for predicting mortality risk in these patients. Harrison [40] identified BNP as a strong predictor of cardiovascular mortality in end-stage renal disease (ESRD) patients, with elevated BNP levels linked to an increased risk of cardiovascular death. It has also been reported that elevated BNP and NT-proBNP levels are associated with an increased risk of all-cause mortality in CKD patients, further reinforcing the role of BNP as a key predictor [41]. These results underscore the importance of monitoring BNP levels and assessing echocardiographic parameters for risk stratification in dialysis patients in clinical practice.

Further, the Cox regression analysis identified LVIDs, age, high-sensitivity Hs-CRP, and CAD as independent predictors of mortality. Specifically, abnormal LVIDs reflect left ventricular dilation and dysfunction caused by volume overload and hypertension, which are commonly observed in dialysis patients. These changes are closely linked to the occurrence of heart failure and arrhythmias. These changes are primarily driven by cardiac stress resulting from volume overload and hypertension. Previous studies have reported that increased LVIDs are significantly correlated with a heightened risk of adverse cardiovascular events in these populations [42]. Furthermore, LVID measurements are widely utilized to predict cardiovascular outcomes in dialysis patients, including heart failure and arrhythmias. The enlargement of LVIDs often results from long-term dialysis effects and persistent volume overload, leading to left ventricular hypertrophy and eventual dilation. A meta-analysis involving ESRD patients demonstrated that larger LVIDs are significant predictors of heart failure and overall mortality [43]. The modality of dialysis (e.g., hemodialysis versus peritoneal dialysis) may also influence LVIDs. Studies indicate that patients undergoing hemodialysis often exhibit more pronounced left ventricular dilation compared to those on peritoneal dialysis. This difference may be attributed to significant fluctuations in fluid balance and intravascular volume during hemodialysis sessions [44]. Monitoring and evaluating LVIDs are essential for guiding therapeutic strategies in dialysis patients. The early detection of significant increases in LVIDs enables timely interventions aimed at controlling fluid overload, potentially mitigating the risk of heart failure. Some studies suggest that optimizing fluid management can prevent or slow the progression of left ventricular dilation in these patients [45]. Thus, regular monitoring of LVIDs is essential for the management of dialysis patients.

The stratified Cox regression analysis evaluated the association between echocardiographic parameters and mortality in hemodialysis patients (Table 5) using three sequential models. These models incorporated cardiovascular history (e.g., CAD, hypertension, and heart failure) and demographic/nutritional factors (e.g., age and albumin) for a comprehensive risk assessment. Abnormal LVIDs remained a significant independent predictor across all the models (HR = 2.66, *p* = 0.021 in Model 1; *p* = 0.019 in Model 3). Female sex was identified as a protective factor, with lower mortality risk compared to males (*p* < 0.05). After further adjustments, left ventricular end-diastolic diameter (LVIDd) and ejection fraction (EF) also emerged as significant predictors, highlighting their relevance in specific clinical contexts. LVID abnormalities reflect severe ventricular dilation and dysfunction caused by volume overload and hypertension, conditions that are frequently observed in ESRD patients [39]. The observed sex differences in mortality risk are likely related to hormonal and physiological factors, with females often showing better survival outcomes [46]. Regular echocardiographic evaluation is essential for monitoring and addressing changes in left ventricular structure and function, with the potential to improve long-term prognosis [47].

This study evaluated the changes in BNP levels and echocardiographic parameters in maintenance hemodialysis patients before and after COVID-19 infection, with a particular focus on the impact of vaccination. The results showed that, in those patients who were not vaccinated, both the maximum (BNP Max) and mean (BNP Mean) BNP levels significantly increased after infection (*p* < 0.001) (Table 6, Figure 2). Additionally, their echocardiographic parameters, such as interventricular septal thickness (IVSD) and left ventricular posterior wall thickness (LVPWD), worsened significantly (*p* < 0.001 and *p* = 0.001, respectively). In contrast, the vaccinated patients did not show significant changes in BNP levels (BNP Max Wilcoxon test *p* = 0.3, z-value analysis *p* = 0.006; BNP Mean Wilcoxon test *p* = 0.098, z-value analysis *p* = 0.02), and no significant changes were observed in the echocardiographic parameters (all *p* > 0.05) (Table 7, Figure 3).

The significant increase in BNP in the unvaccinated patients post-infection reflects worsened myocardial workload and volume overload due to COVID-19 infection, leading to heart function deterioration. On the other hand, the absence of significant BNP changes in the vaccinated patients suggests that vaccination may help to reduce the severity of infection, decrease cardiac load, and lower the risk of heart failure, thus maintaining a relatively stable BNP level. These findings align with previous studies indicating that COVID-19 infection is closely linked to changes in BNP levels, particularly in severe cases where elevated BNP levels are associated with heart failure or cardiac injury [48]. Hemodialysis patients, in particular, face higher risks after COVID-19 infection, including increased cardiac load and organ failure, with significantly higher mortality and hospitalization rates [49]. Regarding vaccination, research shows that COVID-19 vaccines can reduce severe disease and mortality rates in dialysis patients, although some may exhibit lower immune responses, requiring additional monitoring and potential dose adjustments [50].

COVID-19 vaccines may protect heart function through several mechanisms. First, vaccination can mitigate the body’s immune response, particularly by reducing excessive immune reactions and the release of inflammatory cytokines, thus alleviating myocardial damage. This enables quicker clearance of the virus, shortening the course of the infection and reducing the sustained increase in viral load, thus aiding in preventing long-term damage to the heart and vascular system [51,52]. Additionally, in vaccinated patients, the vaccine may help to control volume and hemodynamics, reducing the extra burden on the heart and thereby minimizing further cardiac injury [53,54].

Cardiovascular events play a crucial role in the mortality and complications of maintenance hemodialysis patients. This study suggests that vaccination not only reduces the overall risk of COVID-19 infection or mitigates its severity but may also help to protect cardiac function, preventing or delaying myocardial structural and functional damage caused by the infection. However, while the findings indicate potential cardiovascular protection from vaccination, it is important to note that the effect of the vaccine may be influenced by factors such as patients’ baseline health status. In this study, the vaccinated group was younger and had a lower incidence of CVD, which could contribute to the observed improvement in their BNP levels and echocardiographic parameters. Therefore, the observed improvements may reflect the healthier baseline characteristics of the vaccinated group rather than the direct effect of the vaccine itself. Additionally, the immune-modulating effect of the CoronaVac vaccine in reducing inflammation and improving heart function remains unclear, and the existing literature does not provide strong support for this mechanism, requiring further investigation. Based on these findings, it is recommended that clinical practice actively promote vaccination among hemodialysis patients, particularly high-risk individuals who are unvaccinated or have insufficient vaccine protection. However, the study has limitations, including a small sample size, which may affect the generalizability of the results; a short follow-up period, limiting the ability to assess long-term effects; and a lack of detailed analysis of vaccine type and dosage, which may influence the outcomes. Future large-scale multi-center studies are necessary to further validate and refine these findings.

## 5. Conclusions

In conclusion, this study indicates that COVID-19 vaccination (CoronaVac) is associated with reduced mortality and more stable cardiac function in maintenance hemodialysis patients. The vaccinated patients exhibited smaller changes in their BNP levels and echocardiographic parameters following COVID-19 infection, suggesting that vaccination may enhance survival outcomes by alleviating infection severity and the associated cardiac stress. However, due to the pre-existing prognostic advantages in age and CAD burden in the vaccinated group, further multivariate analysis is required to confirm the independent protective effect of vaccination.

## Figures and Tables

**Figure 1 vaccines-13-00208-f001:**
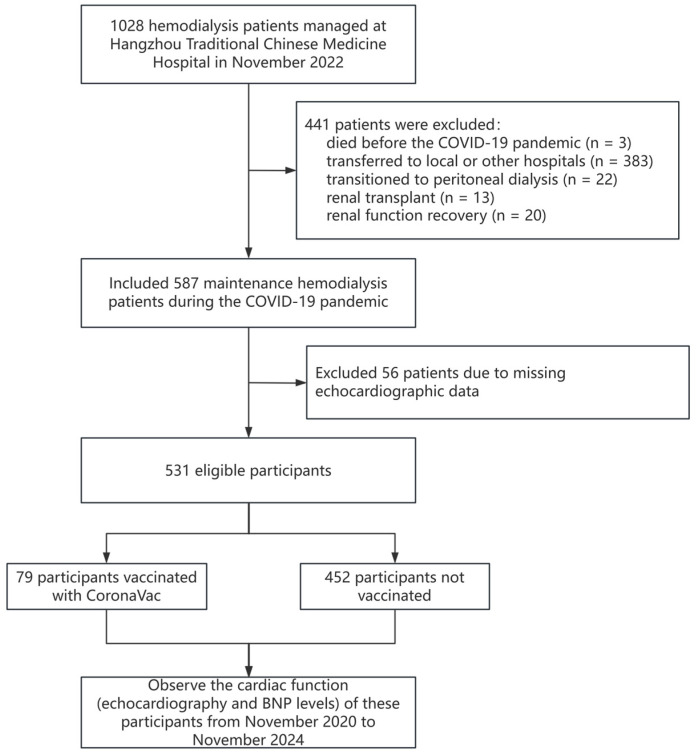
Patient selection flowchart during the COVID-19 pandemic.

**Figure 2 vaccines-13-00208-f002:**
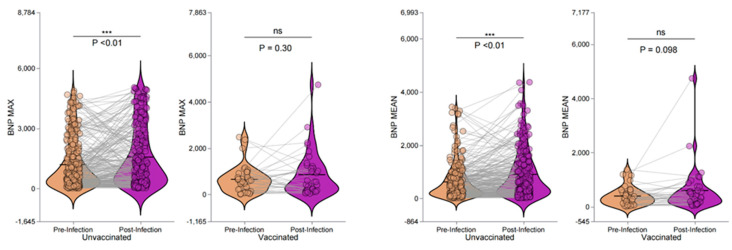
Pre- and post-infection changes in BNP levels in unvaccinated and vaccinated hemodialysis patients. “ns” indicates no significant difference between the groups, “***” indicates a *p*-value less than 0.01.

**Figure 3 vaccines-13-00208-f003:**
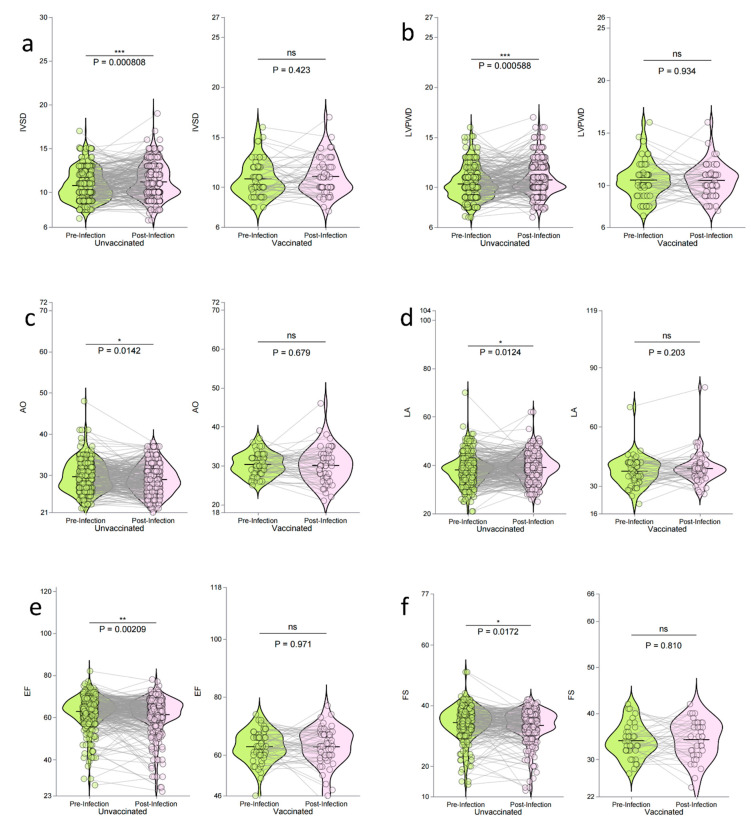
Changes in echocardiographic parameters before and after COVID-19 infection in maintenance hemodialysis patients, comparing vaccinated (CoronaVac) and unvaccinated groups. The panels show changes in interventricular septal thickness (IVSD, **a**), left ventricular posterior wall thickness (LVPWD, **b**), aortic root diameter (AO, **c**), left atrial diameter (LA, **d**), ejection fraction (EF, **e**), and fractional shortening (FS, **f**). Green indicates pre-infection, and purple indicates post-infection. Left panels represent the unvaccinated group, and right panels the vaccinated group. Statistical significance is denoted by * *p* < 0.05, ** *p* < 0.01, *** *p* < 0.001, and ns (not significant).

**Table 1 vaccines-13-00208-t001:** Baseline demographic and clinical characteristics of hemodialysis patients by CoronaVac status.

Variable	Total (n = 531)	Unvaccinated (n = 452)	Vaccinated (n = 79)	t/χ^2^	*p*-Value
Age (years)	64.51 ± 13.81	65.21 ± 13.76	60.54 ± 13.51	t = 2.79	0.006 **
Dialysis Duration (months)	71.10 ± 63.30	73.73 ± 64.79	56.04 ± 51.88	t = 2.30	0.022 *
Gender, n (%)				χ^2^ = 1.18	0.277
Male	334 (62.90)	280 (61.95)	54 (68.35)		
Female	197 (37.10)	172 (38.05)	25 (31.65)		
Primary Disease, n (%)					0.243
PKD	26 (4.90)	20 (4.42)	6 (7.59)		
CTD	9 (1.69)	8 (1.77)	1 (1.27)		
CGN	289 (54.43)	240 (53.10)	49 (62.03)		
DN	199 (37.48)	177 (39.16)	22 (27.85)		
Others	8 (1.51)	7 (1.55)	1 (1.27)		
Death, n (%)				χ^2^ = 3.98	0.049 *
0	461 (86.82)	387 (85.62)	74 (93.67)		
1	70 (13.18)	65 (14.38)	5 (6.33)		
HTN, n (%) *				χ^2^ = 0.01	0.918
0	72 (13.56)	61 (13.50)	11 (13.92)		
1	459 (86.44)	391 (86.50)	68 (86.08)		
Arrhythmia, n (%)				χ^2^ = 3.33	0.068
0	491 (92.47)	414 (91.59)	77 (97.47)		
1	40 (7.53)	38 (8.41)	2 (2.53)		
CAD, n (%)				χ^2^ = 12.37	<0.001 **
0	392 (73.82)	321 (71.02)	71 (89.87)		
1	139 (26.18)	131 (28.98)	8 (10.13)		
HF, n (%)				χ^2^ = 1.48	0.224
0	507 (95.48)	429 (94.91)	78 (98.73)		
1	24 (4.52)	23 (5.09)	1 (1.27)		
COVID-19 Hospitalization, n (%)				χ^2^ = 0.02	0.898
0	452 (85.28)	385 (85.37)	67 (84.81)		
1	78 (14.72)	66 (14.63)	12 (15.19)		
Severe COVID-19, n (%)				χ^2^ = 0.20	0.658
0	512 (96.42)	437 (96.68)	75 (94.94)		
1	19 (3.58)	15 (3.32)	4 (5.06)		

Abbreviations: PKD: polycystic kidney disease; CTD: connective tissue disease; CGN: chronic glomerulonephritis; DN: diabetic nephropathy; HTN: hypertension; CAD: coronary artery disease; HF: heart failure. * *p* < 0.05, ** *p* < 0.01.

**Table 2 vaccines-13-00208-t002:** Comparison of clinical and biochemical characteristics between survival and death groups of hemodialysis patients.

Variable, Unit	Total (n = 531)	Survival (n = 461)	Death (n = 70)	t/Z	*p*-Value
Age, years	64.51 ± 13.81	62.82 ± 13.24	75.67 ± 12.27	t = −7.64	<0.001 **
Dialysis Duration, months	48.00 (26.00, 96.50)	49.00 (27.00, 97.00)	45.38 (18.66, 94.21)	Z = −1.05	0.292
URR, %	68.55 ± 30.37	68.57 ± 32.12	68.37 ± 8.49	t = 0.04	0.97
ALB, g/dL	35.62 ± 4.00	35.96 ± 3.88	33.60 ± 4.15	t = 4.42	<0.001 **
ALT, U/L	14.07 ± 8.47	14.11 ± 8.18	13.81 ± 10.06	t = 0.26	0.794
LDL-C, mg/dL	2.19 ± 0.82	2.21 ± 0.84	2.06 ± 0.68	t = 1.35	0.178
Ca, mg/dL	2.22 ± 0.20	2.22 ± 0.20	2.20 ± 0.17	t = 0.86	0.389
TG, mg/dL	1.84 ± 1.48	1.86 ± 1.53	1.71 ± 1.11	t = 0.73	0.468
HDL-C, mg/dL	0.97 ± 0.24	0.97 ± 0.24	0.92 ± 0.25	t = 1.64	0.102
Cr, µmol/L	750.93 ± 275.31	774.00 ± 276.61	612.10 ± 223.19	t = 5.14	<0.001 **
PTH, pg/mL	226.10 (117.12, 437.43)	224.40 (117.85, 418.15)	273.90 (108.10, 504.00)	Z = −0.65	0.514
K, mEq/L	4.66 ± 0.66	4.68 ± 0.66	4.56 ± 0.65	t = 1.40	0.162
PHOS, mg/dL	1.61 ± 0.46	1.64 ± 0.46	1.42 ± 0.46	t = 3.66	<0.001 **
Na, mEq/L	138.03 ± 3.16	138.12 ± 3.18	137.43 ± 2.96	t = 1.70	0.090 *
Urea, µmol/L	21.80 ± 6.52	22.20 ± 6.25	19.40 ± 7.56	t = 3.19	0.002 **
PT, seconds	11.30 ± 2.02	11.20 ± 2.06	11.89 ± 1.68	t = −2.42	0.016 *
PA, mg/dL	313.06 ± 80.97	318.20 ± 78.71	275.05 ± 88.09	t = 3.25	0.001 **
AST, U/L	18.86 ± 8.75	18.27 ± 7.79	22.24 ± 12.50	t = −2.44	0.017 *
Hb, g/dL	109.90 ± 13.72	110.68 ± 13.38	104.77 ± 14.89	t = 3.37	<0.001 **
UA, µmol/L	401.74 ± 102.71	410.32 ± 98.65	350.48 ± 112.04	t = 4.37	<0.001 **
TCH, mg/dL	3.89 ± 1.11	3.93 ± 1.13	3.63 ± 0.95	t = 1.96	0.051
Hs-CRP, mg/L	2.44 (1.11, 7.32)	2.03 (0.97, 6.20)	7.26 (3.17, 24.80)	Z = −6.30	<0.001 **
Ferritin, ng/mL	80.75 (39.30, 160.82)	74.30 (39.30, 157.10)	120.70 (47.10, 221.90)	Z = −1.88	0.06

Abbreviations: URR: urea reduction ratio; ALB: albumin; ALT: alanine transaminase; LDL-C: low-density lipoprotein cholesterol; Ca: calcium; TG: triglycerides; HDL-C: high-density lipoprotein cholesterol; Cr: creatinine; PTH: parathyroid hormone; K: potassium; PHOS: phosphate; Na: sodium; Urea: urea; PT: prothrombin time; PA: prealbumin; AST: aspartate transaminase; Hb: hemoglobin; UA: uric acid; TCH: total cholesterol; Hs-CRP: high-sensitivity C-reactive protein. * *p* < 0.05, ** *p* < 0.01.

**Table 3 vaccines-13-00208-t003:** Echocardiographic parameters and BNP levels in relation to mortality among maintenance hemodialysis patients.

Variable, Unit	Total (n = 531)	Survival (n = 461)	Death (n = 70)	Z/χ^2^	*p*-Value
BNP Max, pg/mL	462.50 (98.75, 1216.50)	401.50 (86.75, 999.75)	1246.00 (387.75, 2456.50)	Z = −4.77	<0.001 **
BNP Mean, pg/mL	283.00 (84.75, 679.75)	247.00 (70.75, 612.25)	584.50 (181.50, 992.00)	Z = −3.52	<0.001 **
IVSD, mm	10.20 (9.00, 12.00)	10.00 (9.00, 12.00)	11.00 (10.00, 12.00)	Z = −1.39	0.165
LVIDd, mm	49.00 (46.00, 53.00)	49.00 (46.00, 53.00)	49.00 (46.02, 53.00)	Z = −0.08	0.938
LVIDs, mm	32.00 (30.00, 35.00)	32.00 (30.00, 35.00)	33.00 (30.00, 36.40)	Z = −0.99	0.321
LVPWD, mm	10.00 (9.00, 11.00)	10.00 (9.00, 11.00)	10.00 (9.57, 11.65)	Z = −1.53	0.127
AO, mm	29.00 (27.00, 32.00)	29.00 (27.00, 32.00)	29.00 (27.00, 32.00)	Z = −0.36	0.718
LA, mm	38.00 (34.00, 41.00)	38.00 (34.00, 41.00)	39.00 (34.25, 42.00)	Z = −1.14	0.254
SV, mL	72.00 (63.50, 85.00)	72.00 (63.00, 85.00)	73.00 (65.00, 81.00)	Z = −0.12	0.906
EF, %	64.00 (60.00, 68.00)	64.00 (60.00, 68.00)	64.00 (58.00, 66.75)	Z = −1.46	0.144
FS, %	35.00 (32.00, 38.00)	35.00 (32.00, 38.00)	35.00 (31.25, 37.00)	Z = −1.04	0.299
Arrhythmia				χ^2^ = 3.28	0.07
Normal	491 (92.47)	430 (93.28)	61 (87.14)		
Abnormal	40 (7.53)	31 (6.72)	9 (12.86)		
Coronary Artery Disease (CAD)				χ^2^ = 70.02	<0.001 **
Normal	392 (73.82)	369 (80.04)	23 (32.86)		
Abnormal	139 (26.18)	92 (19.96)	47 (67.14)		
Heart Failure				χ^2^ = 7.17	0.007 **
Normal	507 (95.48)	445 (96.53)	62 (88.57)		
Abnormal	24 (4.52)	16 (3.47)	8 (11.43)		
IVSD Status				χ^2^ = 1.07	0.301
Normal	212 (39.92)	188 (40.78)	24 (34.29)		
Abnormal	319 (60.08)	273 (59.22)	46 (65.71)		
LVIDd Status				χ^2^ = 1.50	0.221
Normal	464 (87.38)	406 (88.07)	58 (82.86)		
Abnormal	67 (12.62)	55 (11.93)	12 (17.14)		
LVIDs Status				χ^2^ = 3.86	0.049 *
Normal	228 (90.48)	200 (92.17)	28 (80.00)		
Abnormal	24 (9.52)	17 (7.83)	7 (20.00)		
LVPWDStatus				χ^2^ = 1.71	0.191
Normal	251 (47.27)	223 (48.37)	28 (40.00)		
Abnormal	280 (52.73)	238 (51.63)	42 (60.00)		
AO Status				χ^2^ = 0.00	1
Normal	517 (97.36)	449 (97.40)	68 (97.14)		
Abnormal	14 (2.64)	12 (2.60)	2 (2.86)		
LA Status				χ^2^ = 0.67	0.412
Normal	385 (72.64)	337 (73.26)	48 (68.57)		
Abnormal	145 (27.36)	123 (26.74)	22 (31.43)		
SV Status				χ^2^ = 0.04	0.846
Normal	355 (90.79)	306 (90.53)	49 (92.45)		
Abnormal	36 (9.21)	32 (9.47)	4 (7.55)		
EF Status				χ^2^ = 1.18	0.278
Normal	486 (91.87)	424 (92.37)	62 (88.57)		
Abnormal	43 (8.13)	35 (7.63)	8 (11.43)		
FS Status				χ^2^ = 0.30	0.586
Normal	381 (93.38)	332 (93.79)	49 (90.74)		
Abnormal	27 (6.62)	22 (6.21)	5 (9.26)		

Abbreviations: BNP Max: maximum brain natriuretic peptide; BNP Mean: mean brain natriuretic peptide; IVSD: interventricular septal thickness in diastole; LVIDd: left ventricular internal diameter in diastole; LVIDs: left ventricular internal diameter in systole; LVPWD: left ventricular posterior wall thickness in diastole; AO: aortic root diameter; LA: left atrium diameter; SV: stroke volume; EF: ejection fraction; FS: fractional shortening; Arrhythmia: abnormal heart rhythm. * *p* < 0.05, ** *p* < 0.01.

**Table 5 vaccines-13-00208-t005:** Multivariate Cox regression analysis of echocardiographic and clinical parameters across different models.

Variable	Model 1 HR (95% CI), *p*	Model 2 HR (95% CI), *p*	Model 3 HR (95% CI), *p*
LVIDd	1.46 (0.78–2.71), 0.235	1.03 (0.28–3.76), 0.959	2.85 (0.18–45.72), 0.460
LVPWD	1.36 (0.84–2.19), 0.207	1.19 (0.25–5.68), 0.824	1.58 (0.05–48.08), 0.793
AO	1.06 (0.26–4.33), 0.934	0.00 (0.00–Inf), 0.997	0.00 (0.00–Inf), 0.999
LA	1.25 (0.75–2.06), 0.394	1.17 (0.29–4.65), 0.828	0.12 (0.01–1.96), 0.135
IVSD	1.28 (0.78–2.10), 0.326	5.94 (1.04–33.77), 0.045 *	49.88 (0.85–2938.25), 0.060 *
SV	0.81 (0.29–2.23), 0.679	1.12 (0.25–5.14), 0.879	0.41 (0.02–10.64), 0.593
EF	1.49 (0.71–3.11), 0.289	0.10 (0.01–1.14), 0.064	0.00 (0.00–74.96), 0.259
FS	1.49 (0.59–3.73), 0.398	0.57 (0.05–6.22), 0.649	97.62 (0.02–562,248.02), 0.300
LVIDs	2.66 (1.16–6.10), 0.021 *	28.88 (3.47–240.13), 0.002 **	1574.03 (3.30–750,000.88), 0.019 *
Gender (Female)	1.00 (0.61–1.62), 0.988	0.15 (0.03–0.66), 0.013 *	0.01 (0.00–0.34), 0.011 *
LVIDd (Abnormal)	0.99 (0.96–1.03), 0.779	0.86 (0.74–0.99), 0.046 *	0.78 (0.57–1.08), 0.139
LVIDs (Abnormal)	1.02 (0.97–1.08), 0.434	0.87 (0.75–1.01), 0.074	0.72 (0.51–0.99), 0.050
SV (Abnormal)	1.00 (0.99–1.01), 0.914	1.04 (1.01–1.07), 0.037 *	1.08 (1.01–1.16), 0.023 *
EF (Abnormal)	0.98 (0.95–1.00), 0.089	0.87 (0.76–0.98), 0.027 *	0.71 (0.52–0.96), 0.025 *
AO (Abnormal)	1.00 (0.94–1.07), 0.888	0.93 (0.81–1.06), 0.288	0.65 (0.45–0.95), 0.027 *

Abbreviations: HR: hazard ratio; CI: confidence interval; Model 1: crude analysis; Model 2: adjusted for HTN, arrhythmia, CAD, and HF; Model 3: adjusted for age, HTN, arrhythmia, CAD, HF, ALB, and PA. * *p* < 0.05, ** *p* < 0.01.

**Table 6 vaccines-13-00208-t006:** Changes in BNP levels pre- and post-infection in unvaccinated and vaccinated hemodialysis patients.

Group	Pre-Infection (n = 283)	Post-Infection (n = 283)	z	*p*-Value	z-Value	z *p*-Value
BNP Max, pg/mL						
Unvaccinated	652.00 (234.00, 1765.50)	1059.00 (312.00, 2486.50)	10,629.5	<0.001 **	−1.76	0.078
Vaccinated	573.50 (205.50, 882.50)	489.50 (161.25, 1152.75)	266	0.3	−2.78	0.006 **
BNP Mean, pg/mL						
Unvaccinated	398.00 (158.50, 833.00)	619.00 (226.50, 1291.50)	8658	<0.001 **	−1.56	0.119
Vaccinated	311.00 (121.50, 535.25)	389.50 (155.75, 640.50)	227	0.098	−2.33	0.02 *

* *p* < 0.05, ** *p* < 0.01.

**Table 7 vaccines-13-00208-t007:** Changes in echocardiographic parameters before and after COVID-19 infection in vaccinated and unvaccinated maintenance hemodialysis patients.

	Group	Pre-Infection (n = 348)	Post-Infection (n = 348)	t-Value	*p*-Value
IVSD, mm	Unvaccinated	10.76 ± 1.73	11.18 ± 1.95	−3.39	<0.001 **
	Vaccinated	10.84 ± 1.81	11.07 ± 1.86	−0.81	0.423
LVIDd, mm	Unvaccinated	49.48 ± 5.65	49.79 ± 5.33	−0.82	0.411
	Vaccinated	49.71 ± 5.64	49.59 ± 5.35	0.13	0.897
LVIDS, mm	Unvaccinated	33.52 ± 6.01	33.61 ± 5.81	0.15	0.883
	Vaccinated	32.80 ± 3.36	32.23 ± 4.30	1.3	0.206
LVPWD, mm	Unvaccinated	10.34 ± 1.58	10.96 ± 3.12	−3.2	0.001 **
	Vaccinated	10.50 ± 1.78	10.48 ± 1.57	0.08	0.934
AO, mm	Unvaccinated	29.68 ± 3.73	28.92 ± 3.95	2.62	0.014 *
	Vaccinated	30.35 ± 2.83	29.53 ± 5.98	0.95	0.679
LA, mm	Unvaccinated	38.15 ± 6.06	39.20 ± 5.73	−2.51	0.013 *
	Vaccinated	37.58 ± 7.33	38.91 ± 7.82	−1.29	0.203
SV, mL	Unvaccinated	74.95 ± 16.79	75.99 ± 17.05	−0.35	0.727
	Vaccinated	81.57 ± 32.12	78.55 ± 13.44	0.63	0.531
EF, %	Unvaccinated	62.89 ± 7.64	61.39 ± 8.59	3.11	0.002 **
	Vaccinated	63.30 ± 5.08	63.13 ± 6.27	0.27	0.971
FS, %	Unvaccinated	34.37 ± 5.44	33.32 ± 5.58	2.37	0.017 *
	Vaccinated	34.11 ± 3.72	34.44 ± 4.38	−0.24	0.810

Abbreviations: Data are presented as mean ± standard deviation. IVSD: interventricular septal thickness in diastole; LVIDd: left ventricular internal diameter in diastole; LVIDS: left ventricular internal diameter in systole; LVPWD: left ventricular posterior wall thickness in diastole; AO: aortic root diameter; LA: left atrium diameter; SV: stroke volume; EF: ejection fraction; FS: fractional shortening. * *p* < 0.05, ** *p* < 0.01.

## Data Availability

The data supporting the reported results can be obtained upon reasonable request.

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
