# Peer review of "Impact of COVID-19 Vaccination on Cardiac Function and Survival in Maintenance Hemodialysis Patients"

_vaccines, 2025, doi:10.3390/vaccines13030208_

Round 1

Reviewer 1 Report

Comments and Suggestions for Authors

Regarding the paper by Xiao Tu et al. entitled "Impact of COVID-19 vaccination on cardiac function and survival in patients on maintenance hemodialysis", I have several concerns.

Main ones are

1. What could CoronaVac have that could explain the results of this study?

Minor

1. Please state the meaning of each acronym the first time it appears.

2.- The authors should explain the limitations of the study. Why CoronaVac and not other similar vaccines? Also, the use of multiple comparisons without adjustment for multiplicity could increase the risk of type I error.

Author Response

Reviewer1

Comments 1: 1. What could CoronaVac have that could explain the results of this study?

Response 1:Thank you for raising this question. I/We agree with this comment. Therefore, I/We have made the following revisions in the revised manuscript (see lines 482-503). In this section, we discuss the importance of cardiovascular events in the mortality and complications of maintenance hemodialysis patients. This study suggests that vaccination not only reduces the risk of COVID-19 infection or mitigates its severity but may also help protect cardiac function and reduce myocardial damage caused by the infection. However, the effect of the vaccine may be influenced by patients' baseline health conditions, as the vaccinated group was younger and had a lower incidence of cardiovascular disease, which may explain the observed improvements in BNP and echocardiographic parameters. Additionally, the immune-modulating effect of the CoronaVac vaccine in reducing inflammation and improving heart function remains unclear, and existing literature does not provide strong support for this mechanism, requiring further investigation.

Comments 2:Please state the meaning of each acronym the first time it appears.

Response2:Agree. I/We have, accordingly, revised the manuscript to state the meaning of each acronym the first time it appears. This change was made to ensure clarity and avoid ambiguity for readers who may not be familiar with the terms. Specifically, we have expanded the acronyms such as "BNP" (B-type natriuretic peptide) and "CVD" (cardiovascular disease) at their first occurrence in the manuscript.

Comments 3:The authors should explain the limitations of the study. Why CoronaVac and not other similar vaccines? Also, the use of multiple comparisons without adjustment for multiplicity could increase the risk of type I error.

Response 3:

Thank you for your insightful comments. Due to the circumstances at the time, only the CoronaVac vaccine was available for use. Additionally, as hemodialysis patients represent a special population, a large portion of this group did not have access to or choose to receive the vaccine. We appreciate the reviewer’s concern regarding multiple comparisons and the risk of Type I error.

In this study, we primarily used multivariable Cox regression analysis, which incorporates multiple covariates and adjusts for them simultaneously. This approach inherently controls some of the effects caused by multiple comparisons, thus reducing the risk of Type I error. Moreover, the main objective of this study was to assess the impact of various clinical factors on survival rates, rather than selecting significant variables from a larger set. As a result, we did not use more stringent correction methods, such as Bonferroni adjustment, to avoid overcorrection and the potential risk of Type II error (false negatives).However, to further minimize the risk of Type I error, we applied the False Discovery Rate (FDR) correction (Benjamini-Hochberg) in the analysis of biochemical indicators and echocardiographic parameters to adjust for the impact of multiple hypothesis testing. Sensitivity analysis showed that the key study conclusions remained statistically significant.

Reviewer 2 Report

Comments and Suggestions for Authors

The topic of the paper is of great interest. The authors analyse the impact of COVID-19 vaccination on cardiac function in dialysis patients. The importance of vaccination in this high-risk population is of great importance and has a great deal of clinical practice. It is a novel and well-planned study. Publications on this topic are limited.

The abstract is dense and sometimes confusing. It should be clarified and simplified. In the abstract, the use of acronyms should be limited to facilitate reading.

The introduction and material and methods sections are well written.

However, the information presented in the results section is excessive, complex and confusing and should focus on the objectives of the work: vaccination, cardiac function and haemodialysis. The work is difficult to follow. The authors in lines 297 (results) and 466 (discussion) highlight the objective of the study and this idea is what should be present throughout the work. Therefore, they should focus on this information and simplify the verbose and excessive information they include. This also occurs in the discussion section, where the authors are scattered through excessive information and sometimes comment on and discuss logical results. Reading and following a scientific work should not be drowned in data and information.

Finally, as the authors comment, age and CAD burden in the vaccinated group have a great importance in the results obtained, so the question arises to what extent it is not an important bias in the results obtained.

References must be taken care of and there are important errors in the list.

Author Response

Comments1:The abstract is dense and sometimes confusing. It should be clarified and simplified. In the abstract, the use of acronyms should be limited to facilitate reading.

Response1:Thank you for your feedback. We have revised the abstract to improve clarity and simplicity, limiting the use of acronyms to enhance readability. The revised abstract now provides a more concise overview of the study, focusing on the key findings and their implications. The main conclusions remain the same, emphasizing the positive impact of the COVID-19 vaccine (CoronaVac) on mortality and cardiac function in maintenance hemodialysis patients. You can refer to lines 25-44 for the updated abstract in the manuscript

Comments2:However, the information presented in the results section is excessive, complex and confusing and should focus on the objectives of the work: vaccination, cardiac function and haemodialysis. The work is difficult to follow. The authors in lines 297 (results) and 466 (discussion) highlight the objective of the study and this idea is what should be present throughout the work. Therefore, they should focus on this information and simplify the verbose and excessive information they include. This also occurs in the discussion section, where the authors are scattered through excessive information and sometimes comment on and discuss logical results. Reading and following a scientific work should not be drowned in data and information.

Response2:Thank you for your careful review and valuable feedback. In the results and discussion sections, we have streamlined the language to highlight the core information and avoid unnecessary repetition and redundancy. Given the complexity of the study, particularly with regards to vaccination, cardiac function, and hemodialysis, necessary details have been included.We have revised lengthy sentences in both the results and discussion sections to improve readability and clarity.We have clearly aligned both sections with the primary objective of assessing the impact of vaccination on cardiac function in dialysis patients and how cardiac parameters and BNP levels predict mortality. Non-essential details have been minimized or removed.In the discussion, we have focused on explaining how changes in cardiac function help interpret the protective effect of vaccination on dialysis patients, removing excessive discussions on unrelated factors to create a more concise structure.

Comments3:Finally, as the authors comment, age and CAD burden in the vaccinated group have a great importance in the results obtained, so the question arises to what extent it is not an important bias in the results obtained.

Response3:Thank you for your valuable comments. In lines 466-486, we have addressed the potential bias related to age and CAD burden in the vaccinated group. We noted that the vaccinated group was younger and had a lower incidence of cardiovascular disease, which could explain the observed improvements in BNP and echocardiographic parameters, rather than the direct effect of the vaccine itself. Additionally, the immune-modulating effect of the CoronaVac vaccine on inflammation and heart function requires further investigation.

Comments4:References must be taken care of and there are important errors in the list.

Response4:Thank you for pointing out the issue with the references. Due to a technical problem with the reference management software, some references did not display correctly. We have revised and corrected references 6 and 10. The updated references have now been accurately included in the list. We apologize for any inconvenience caused and appreciate your understanding.

Reviewer 3 Report

Comments and Suggestions for Authors

Comments and suggestions

This study aims to evaluate the effect of COVID-19 vaccination (CoronaVac) on cardiac function and survival in this population.

Summary section:

1. Results and conclusions must be aligned

Introduction section

2. Go deeper into the topic of maintenance hemodialysis

3. Describe the severity of COVID-19 with cardiac function

Methodology section:

4. Delimit the inclusion and exclusion criteria

5. Why was COVID-19 not considered to be stratified into mild, moderate and severe.

Results section:

6. Restructure the second paragraph for better understanding.

References section:

7. Check the relevance of the references and change or clarify what happened with reference number 6.

Author Response

Comments 1 Results and conclusions must be aligned

Response1:Thank you for your comment. We have revised the manuscript to ensure that the results and conclusions are more closely aligned. Specifically, in lines 35-45, we have updated the text to clarify how the results support the conclusions drawn, emphasizing the relationship between vaccination and the observed outcomes in the study population.

Comments 2: Go deeper into the topic of maintenance hemodialysis

Response2:Thank you for your comment. We have expanded the discussion on maintenance hemodialysis in the manuscript to provide a deeper understanding of this critical topic. Specifically, in the introduction section (L88-109), we have elaborated on the challenges faced by patients on maintenance hemodialysis, including the impact on cardiovascular health, increased infection risk, and the effect of dialysis on overall mortality.

Comments 3:. Describe the severity of COVID-19 with cardiac function

Response3:Thank you for your comment. We have provided a detailed description of the severe impact of COVID-19 on cardiac function in lines 75-85 of the manuscript. This section discusses how COVID-19 infection can lead to myocardial injury, impaired cardiac function, and potentially exacerbate pre-existing cardiovascular conditions, thereby affecting patient prognosis.

Comments 4:. Delimit the inclusion and exclusion criteria

Response 4:Thank you for your comment. We have revised the inclusion and exclusion criteria for better clarity. The updated details can be found in lines 127-140 of the revised manuscript. The revised section now clearly delineates the specific criteria used to include or exclude patients from the study, ensuring a more transparent and reproducible methodology.

Comments5:. Why was COVID-19 not considered to be stratified into mild, moderate and severe.

Response 5:Thank you for your comment. Due to the absence of clear classification standards for COVID-19 severity at the time of the study, we were unable to stratify the patients into mild, moderate, and severe categories. Most patients were infected within a one-month period, and many laboratory tests could not be completed or were incomplete during that time. As a result, we classified patients based on their hospitalization status and mortality outcomes, which were the most reliable indicators available at the time.

Comments6: Restructure the second paragraph for better understanding.

Response6:Thank you for your suggestion. We have revised the second paragraph to improve clarity and flow. The updated text can be found in lines 181-186. We restructured the sentences to more clearly convey the significantly lower mortality rate in the vaccinated group and to highlight the independent mortality predictors identified in the multivariate analysis, including age, coronary artery disease, inflammation levels, and abnormal left ventricular systolic diameter. The revision improves the overall readability and understanding of the paragraph.

Comments7: Check the relevance of the references and change or clarify what happened with reference number 6.

Response7:Thank you for your comment. We have carefully reviewed the relevance of all the references in the manuscript and made the necessary updates. Regarding reference number 6, due to an issue with the EndNote version, it was not displayed correctly in the previous version. This has now been addressed and the correction has been made in L549-550.

Round 2

Reviewer 1 Report

Comments and Suggestions for Authors

The authors have answered all my concerns.